# WHICH MUTUAL-INFORMATION REPRESENTATION LEARNING OBJECTIVES ARE SUFFICIENT FOR CONTROL?

## ABSTRACT

Mutual information maximization provides an appealing formalism for learning representations of data. In the context of reinforcement learning, such representations can accelerate learning by discarding irrelevant and redundant information, while retaining the information necessary for control. Much of the prior work on these methods has addressed the practical difficulties of estimating mutual information from samples of high-dimensional observations, while comparatively less is understood about *which* mutual information objectives are sufficient for reinforcement learning (RL) from a theoretical perspective. In this paper we identify conditions under which representations that maximize specific mutual-information objectives are theoretically sufficient for learning and representing the optimal policy. Somewhat surprisingly, we find that several popular objectives can yield insufficient representations given mild and common assumptions on the structure of the MDP. We corroborate our theoretical results with empirical results experiments on a simulated game environment with visual observations.

## 1   INTRODUCTION

While deep reinforcement learning (RL) algorithms are capable of learning policies from high-dimensional observations, such as images (Mnih et al., 2013; Lee et al., 2019; Kalashnikov et al., 2018), in practice policy learning faces a bottleneck in acquiring useful representations of the observation space (Shelhamer et al., 2016). State representation learning approaches aim to remedy this issue by learning structured and compact representations on which to perform RL. While a wide range of representation learning objectives have been proposed (Lesort et al., 2018), a particularly appealing class of methods that is amenable to rigorous analysis is based on maximizing mutual information (MI) between variables. In the unsupervised learning setting, this is often realized as the InfoMax principle (Linsker, 1988; Bell & Sejnowski, 1995), which maximizes the mutual information between the input and its latent representation subject to domain-specific constraints. This approach has been widely applied in unsupervised learning in the domains of image, audio, and natural language understanding (Oord et al., 2018; Hjelm et al., 2018; Ravanelli & Bengio, 2019). In RL, the variables of interest for MI maximization are sequential states, actions, and rewards (see Figure 1). As we will discuss, several popular methods for representation learning in RL involve mutual information maximization with different combinations of these variables (Anand et al., 2019; Oord et al., 2018; Pathak et al., 2017; Shelhamer et al., 2016).

A useful representation should retain the factors of variation that are necessary to learn and represent the optimal policy or the optimal value function, and discard irrelevant and redundant information. While much prior work has focused on the problem of how to optimize various mutual information objectives in high dimensions (Song & Ermon, 2019; Belghazi et al., 2018; Oord et al., 2018; Hjelm et al., 2018), we focus instead on whether the representations that maximize these objectives are actually theoretically sufficient for learning and representing the optimal policy or value function. We find that some commonly used objectives are insufficient given relatively mild and common assumptions on the structure of the MDP, and identify other objectives which are sufficient. We show these results theoretically and illustrate the analysis empirically in didactic examples in which MI can be computed exactly. Our results provide some guidance to the deep RL practitioner on when and why objectives may be expected to work well or fail, and also provide a framework to analyze newly

proposed representation learning objectives based on MI. To investigate how our theoretical results pertain to deep RL, we compare the performance of RL agents in a simulated game trained with state representations learned by maximizing the MI objective given visual inputs. The experimental results corroborate our theoretical findings, and demonstrate that the sufficiency of a representation can have a substantial impact on the performance of an RL agent that uses that representation.

## 2    RELATED WORK

In this paper, we analyze several widely used mutual information objectives for control. In this section we first review MI-based unsupervised learning, then the application of these techniques to the RL setting. Finally, we discuss alternative perspectives on representation learning in RL.

**Mutual information-based unsupervised learning.** Mutual information-based methods are particularly appealing for representation learning as they admit both rigorous analysis and intuitive interpretation. Tracing its roots to the InfoMax principle (Linsker, 1988; Bell & Sejnowski, 1995), a common technique is to maximize the MI between the input and its latent representation subject to domain-specific constraints (Becker & Hinton, 1992). This technique has been applied to learn representations for natural language (Devlin et al., 2019), video (Sun et al., 2019), and images (Bachman et al., 2019; Hjelm et al., 2018). A major challenge to using MI maximization methods in practice is the difficulty of estimating MI from samples (McAllester & Statos, 2018) and with high-dimensional inputs (Song & Ermon, 2019). Much recent work has focused on improving MI estimation via variational methods (Song & Ermon, 2019; Poole et al., 2019; Oord et al., 2018; Belghazi et al., 2018). In this work we are concerned with analyzing the MI objectives, and not the estimation method. In our experiments with image observations, we make use of noise contrastive estimation methods (Gutmann & Hyvärinen, 2010), though other choices could also suffice.

**Mutual information objectives in RL.** Reinforcement learning adds aspects of temporal structure and control to the standard unsupervised learning problem discussed above (see Figure 1). This structure can be leveraged by maximizing MI between sequential states, actions, or combinations thereof. Some works omit the action, maximizing the MI between current and future states (Anand et al., 2019; Oord et al., 2018; Stooke et al., 2020). Much prior work learns latent forward dynamics models (Watter et al., 2015; Karl et al., 2016; Zhang et al., 2018b; Hafner et al., 2019; Lee et al., 2019), related to the forward information objective we introduce in Section 4. Multi-step inverse models, closely related to the inverse information objective (Section 4), have been used to learn control-centric representations (Yu et al., 2019; Gregor et al., 2016). Single-step inverse models have been deployed as regularization of forward models (Zhang et al., 2018a; Agrawal et al., 2016) and as an auxiliary loss for policy gradient RL Shelhamer et al. (2016); Pathak et al. (2017). The MI objectives that we study have also been used as reward bonuses to improve exploration, without impacting the representation, in the form of empowerment (Klyubin et al., 2008; 2005; Mohamed & Rezende, 2015; Leibfried et al., 2019) and information-theoretic curiosity (Still & Precup, 2012).

**Representation learning for reinforcement learning.** In RL, the problem of finding a compact state space has been studied as state aggregation or abstraction (Bean et al., 1987; Li et al., 2006). Abstraction schemes include bisimulation (Givan et al., 2003), homomorphism (Ravindran & Barto, 2003), utile distinction (McCallum, 1996), and policy irrelevance (Jong & Stone, 2005). While efficient algorithms exist for MDPs with known transition models for some abstraction schemes such as bisimulation (Ferns et al., 2006; Givan et al., 2003), in general obtaining error-free abstractions is highly impractical for most problems of interest. For approximate abstractions prior work has bounded the sub-optimality of the policy (Bertsekas et al., 1988; Dean & Givan, 1997; Abel et al., 2016) as well as the sample efficiency (Lattimore & Szepesvari, 2019; Van Roy & Dong, 2019; Du et al., 2019), with some results extending to the deep learning setting (Gelada et al., 2019; Nachum et al., 2018). In this paper, we focus on whether a representation can be used to learn the optimal policy, and not the tractability of learning. Alternatively, priors based on the structure of the physical world can be used to guide representation learning (Jonschkowski & Brock, 2015). In deep RL, many auxiliary objectives distinct from the objectives that we study have been proposed, including meta-learning general value functions (Veeriah et al., 2019), predicting multiple value functions (Bellemare et al., 2019; Fedus et al., 2019; Jaderberg et al., 2016) and predicting domain-specific measurements (Mirowski, 2019; Dosovitskiy & Koltun, 2016). We restrict our analysis to objectives that can be expressed as MI-maximization.

# 3 REPRESENTATION LEARNING FOR RL

The goal of representation learning for RL is to learn a compact representation of the state space that discards irrelevant and redundant information. In this section we formalize each part of this statement, starting with defining the RL problem and representation learning in the context of RL. We then propose and define the metric of sufficiency to evaluate the usefulness of a representation.

## 3.1 PRELIMINARIES

We begin with brief preliminaries of reinforcement learning and mutual information.

**Reinforcement learning.** A Markov decision process (MDP) is defined by the tuple $(\mathcal{S}, \mathcal{A}, \mathcal{T}, r)$, where $\mathcal{S}$ is the set of states, $\mathcal{A}$ the set of actions, $\mathcal{T} : \mathcal{S} \times \mathcal{A} \times \mathcal{S} \to [0, 1]$ the state transition distribution, and $r : \mathcal{S} \times \mathcal{A} \times \mathcal{S} \to \mathbb{R}$ the reward function. We will use capital letters to refer to random variables and lower case letters to refer to values of those variables (e.g., $S$ is the random variable for the state and $\mathbf{s}$ is a specific state). Throughout our analysis we will often be interested in multiple reward functions, and denote a set of reward functions as $\mathcal{R}$. The objective of RL is to find a policy that maximizes the sum of discounted returns $\bar{R}$ for a given reward function $r$, and we denote this optimal policy as $\pi_r^* = \arg\max_\pi \mathbb{E}_\pi[\sum_t \gamma^t r(S_t, A_t)]$ for discount factor $\gamma$. We also define the optimal $Q$-function as $Q_r^*(\mathbf{s}_t, \mathbf{a}_t) = \mathbb{E}_{\pi^*}[\sum_{t=1}^\infty \gamma^t r(S_t, A_t)|\mathbf{s}_t, \mathbf{a}_t]$. The optimal $Q$-function satisfies the recursive Bellman equation, $Q_r^*(\mathbf{s}_t, \mathbf{a}_t) = r(\mathbf{s}_t, \mathbf{a}_t) + \gamma \mathbb{E}_{p(\mathbf{s}_{t+1}|\mathbf{s}_t, \mathbf{a}_t)} \arg\max_{\mathbf{a}_{t+1}} Q_r^*(\mathbf{s}_{t+1}, \mathbf{a}_{t+1})$. The optimal policy and the optimal Q-function are related according to $\pi^*(\mathbf{s}) = \arg\max_{\mathbf{a}} Q^*(\mathbf{s}, \mathbf{a})$.

**Mutual information.** In information theory, the mutual information (MI) between two random variables, $X$ and $Y$, is defined as (Cover, 1999):

$$I(X; Y) = \mathbb{E}_{p(x,y)} \log \frac{p(x, y)}{p(x)p(y)} = H(X) - H(X|Y). \tag{1}$$

The first definition indicates that MI can be understood as a relative entropy (or KL-divergence), while the second underscores the intuitive notion that MI measures the reduction in the uncertainty of one random variable from observing the value of the other.

**Representation learning for RL.** The goal of representation learning for RL is to find a compact representation of the state space that discards details in the state that are not relevant for representing the policy or value function, while preserving task-relevant information (see Figure 1). While state aggregation methods typically define deterministic rules to group states in the representation (Bean et al., 1987; Li et al., 2006), MI-based representation learning methods used for deep RL treat the representation as a random variable (Nachum et al., 2018; Oord et al., 2018; Pathak et al., 2017). Accordingly, we formalize a representation as a stochastic mapping between original state space and representation space.

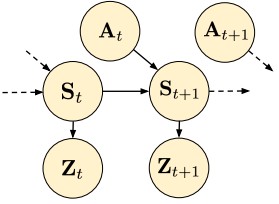

Figure 1: Probabilistic graphical model illustrating the state representation learning problem, estimating state representation $Z$ from original state $S$.

**Definition 1.** *A **stochastic representation** $\phi_{\mathcal{Z}}(\mathbf{s})$ is a mapping from states $\mathbf{s} \in \mathcal{S}$ to a probability distribution $p(Z|S = \mathbf{s})$ over elements of a new representation space $z \in \mathcal{Z}$.*

In this work we consider learning state representations from data by maximizing an objective $\mathbb{J}$. Given an objective $\mathbb{J}$, we define the set of representations that maximize this objective as $\Phi_{\mathbb{J}} = \{\phi_{\mathcal{Z}}\}$ s.t. $\phi_{\mathcal{Z}} \in \arg\max \mathbb{J}(\phi)$.

Unlike problem formulations for partially observed settings (Watter et al., 2015; Hafner et al., 2019; Lee et al., 2019), we assume that $S$ is a Markovian state; therefore the representation for a given state is conditionally independent of the past states, a common assumption in the state aggregation literature (Bean et al., 1987; Li et al., 2006). See Figure 1 for a depiction of the graphical model.

## 3.2 Sufficient Representations for Reinforcement Learning

We now turn to the problem of evaluating stochastic representations for RL. Intuitively, we expect a useful state representation to be capable of representing the optimal policy in the original state space.

**Definition 2.** *A representation $\phi_{\mathcal{Z}}$ is $\pi^*$-**sufficient** with respect to a set of reward functions $\mathcal{R}$ if $\forall r \in \mathcal{R}, \phi_{\mathcal{Z}}(\mathbf{s}_1) = \phi_{\mathcal{Z}}(\mathbf{s}_2) \implies \pi_r^*(A|\mathbf{s}_1) = \pi_r^*(A|\mathbf{s}_2)$.*

When a stochastic representation $\phi_{\mathcal{Z}}$ produces the same distribution over the representation space for two different states $\mathbf{s}_1$ and $\mathbf{s}_2$ we say it *aliases* these states. Unfortunately, as already proven in Theorem 4 of Li et al. (2006) for the more restrictive case of deterministic representations, being able to represent the optimal policy does not guarantee that it can be learned via RL in the representation space. Accordingly, we define a stricter notion of sufficiency that *does* guarantee the convergence of Q-learning to the optimal policy in the original state space (refer to Theorem 4 of Li et al. (2006) for the proof of this).

**Definition 3.** *A representation $\phi_{\mathcal{Z}}$ is $Q^*$-**sufficient** with respect to a set of reward functions $\mathcal{R}$ if $\forall r \in \mathcal{R}, \phi_{\mathcal{Z}}(\mathbf{s}_1) = \phi_{\mathcal{Z}}(\mathbf{s}_2) \implies \forall \mathbf{a}, Q_r^*(\mathbf{a}, \mathbf{s}_1) = Q_r^*(\mathbf{a}, \mathbf{s}_2)$.*

Note that $Q^*$-sufficiency implies $\pi^*$-sufficiency since the optimal policy and the optimal Q-function are directly related via $\pi_r^*(s) = \arg\max_a Q_r^*(s, a)$ (Sutton & Barto, 2018); however the converse is not true. We emphasize that while $Q^*$-sufficiency guarantees convergence, it does not guarantee tractability, which has been explored in prior work (Lattimore & Szepesvari, 2019; Du et al., 2019).

We will further say that an *objective* $\mathbb{J}$ is sufficient with respect to some set of reward functions $\mathcal{R}$ if all the representations that maximize that objective $\Phi_{\mathbb{J}}$ are sufficient with respect to every element of $\mathcal{R}$ according to the definition above. Surprisingly, we will demonstrate that not all commonly used objectives satisfy this basic qualification even when $\mathcal{R}$ contains a single known reward function.

## 4 Mutual Information for Representation Learning in RL

In our study, we consider several MI objectives proposed in the literature.

**Forward information:** A commonly sought characteristic of a state representation is to ensure it retains maximum predictive power over future state representations. This property is satisfied by representations maximizing the following MI objective,

$$\mathbb{J}_{fwd} = I(Z_{t+k}; Z_t, A_t) = H(Z_{t+k}) - H(Z_{t+k}|Z_t, A_t). \tag{2}$$

We suggestively name this objective "forward information" due to the second term, which is the entropy of the forward dynamics distribution. This objective is related to that proposed in Nachum et al. (2018), where they consider a sequence of actions.

**State-only transition information:** Several popular methods (Oord et al., 2018; Anand et al., 2019; Stooke et al., 2020) optimize a similar objective, but do not include the action [1]:

$$\mathbb{J}_{state} = I(Z_{t+k}; Z_t) = H(Z_{t+k}) - H(Z_{t+k}|Z_t). \tag{3}$$

As we will show, the exclusion of the action can have a profound effect on the characteristics of the resulting representations.

**Inverse information:** Another commonly sought characteristic of state representations is to retain maximum predictive power of the action distribution that could have generated an observed transition from $\mathbf{s}_t$ to $\mathbf{s}_{t+1}$. Such representations can be learned by maximizing the following information theoretic objective:

$$\mathbb{J}_{inv} = I(A_t; Z_{t+k}|Z_t) = H(A_t|Z_t) - H(A_t|Z_t, Z_{t+k}) \tag{4}$$

We suggestively name this objective "inverse information" due to the second term, which is the entropy of the inverse dynamics. A wide range of prior work learns representations by optimizing

---

[1]It is common to use a history of states, optimizing an objective like $I(Z_{t+k}; Z_1, ..., Z_t)$. In our setting we assume the given state is Markovian, in which case the objectives are equivalent.

closely related objectives (Gregor et al., 2016; Shelhamer et al., 2016; Agrawal et al., 2016; Pathak et al., 2017; Yu et al., 2019; Zhang et al., 2018a). Intuitively, inverse models allow the representation to capture only the elements of the state that are necessary to predict the action, allowing the discard of potentially irrelevant information.

## 5    SUFFICIENCY ANALYSIS

In this section we analyze the sufficiency for control of representations obtained by maximizing each objective presented in Section 4. To focus on the representation learning problem, we decouple it from RL by assuming access to a dataset of transitions collected with a policy that reaches all states with some probability, which can then be used to learn the desired representation. We also assume that distributions, such as the dynamics or inverse dynamics, can be modeled with arbitrary accuracy, and that the maximizing set of representations for a given objective can be computed. While these assumptions might be relaxed in any practical RL algorithm, and exploration plays a confounding role, studying these objectives under such simplifying assumptions allows us to compare them in terms of sufficiency on an equal playing field, isolating the role of representation learning from other confounding components of a complete RL algorithm.

### 5.1    FORWARD INFORMATION

In this section we show that a representation that maximizes $\mathbb{J}_{fwd}$ is sufficient for optimal control under any reward function. This result aligns with intuition that a representation that captures forward dynamics can represent everything predictable in the state space, and can thus be used to learn the optimal policy for any task. Note that this strength can also be a weakness if there are many predictable elements that are irrelevant for downstream tasks, since the representation retains more information than is needed for the task.

**Proposition 1.** $\mathbb{J}_{fwd}$ *is sufficient for all reward functions.*

*Proof. (Sketch)* We first show that if $Z_t, A_t$ are maximally informative of $Z_{t+k}$, they are also maximally informative of the return $\bar{R}_t$. Due to the Markov structure, $\mathbb{E}_{p(Z_t|S_t=\mathbf{s})}p(\bar{R}_t|Z_t, A_t) = p(\bar{R}_t|S_t = \mathbf{s}, A_t)$. In other words, given $\phi_{\mathcal{Z}}$, additionally knowing $S$ doesn't change our belief about the future return. The $Q$-value is the expectation of the return, so $Z$ has as much information about the $Q$-value as $S$ does. The full proof can be found in Appendix 8.1. $\square$

### 5.2    STATE-ONLY TRANSITION INFORMATION

While $\mathbb{J}_{state}$ is closely related to $\mathbb{J}_{fwd}$, we now show that it is not sufficient.

**Proposition 2.** $\mathbb{J}_{state}$ *is not sufficient for all reward functions.*

*Proof.* Consider the counter-example in Figure 2. Suppose that the two actions $\mathbf{a}_0$ and $\mathbf{a}_1$ are equally likely under the policy distribution. Each state gives no information about which of the two possible next states is more likely; this depends on the action. Therefore, a representation maximizing $\mathbb{J}_{state}$ is free to alias states with the same next-state distribution, such as $\mathbf{s}_0$ and $\mathbf{s}_3$. An alternative view is that such a representation can maximize $\mathbb{J}_{state} = H(Z_{t+k}) - H(Z_{t+k}|Z_t)$ by reducing both terms in equal amounts - aliasing $\mathbf{s}_0$ and $\mathbf{s}_3$ decreases the marginal entropy as well as the entropy of predicting the next state starting from $\mathbf{s}_1$ or $\mathbf{s}_2$. However, this aliased representation is not capable of representing the optimal policy which must distinguish $\mathbf{s}_0$ and $\mathbf{s}_3$ in order to choose the correct action to reach $\mathbf{s}_2$, which yields reward. $\square$

### 5.3    INVERSE INFORMATION

Here we show that representations that maximize $\mathbb{J}_{inv}$ are not sufficient for control in all MDPs. Intuitively, one way that the representation can be insufficient is by retaining only controllable state

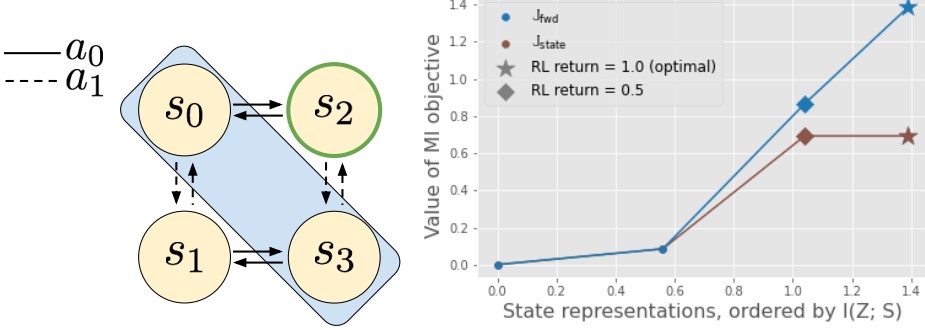

Figure 2: (left) A representation that aliases the states $s_0$ and $s_3$ into a single state maximizes $\mathbb{J}_{state}$ but is not sufficient to represent the optimal policy which must choose different actions in $s_0$ and $s_3$ to reach $s_2$ which yields reward. (right) Values of $\mathbb{J}_{state}$ and $\mathbb{J}_{fwd}$ for a few representative state representations, ordered by increasing $I(Z; S)$. The representation that aliases $s_0$ and $s_3$ (plotted with a diamond) maximizes $\mathbb{J}_{state}$, but the policy learned with this representation may not be optimal (as shown here). The original state representation (plotted with a star) is sufficient.

elements, while the reward function depends on state elements outside the agent's control. We then show that additionally representing the immediate reward is not enough to resolve this issue.

**Proposition 3.** *$\mathbb{J}_{inv}$ is not sufficient for all reward functions. Additionally, adding $I(R_t; Z_t)$ to the objective does not make it sufficient.*

*Proof.* Consider the MDP illustrated in Figure 3, and the representation that aliases the states $s_0$ and $s_1$. The same actions taken from these states lead to different next states which may have different rewards ($a_0$ leads to the reward from $s_0$ while $a_1$ leads to the reward from $s_1$). However, this representation maximizes $\mathbb{J}_{inv}$ because given each pair of states, the action is identifiable. Interestingly, this problem cannot be remedied by simply requiring that the representation also be capable of predicting immediate rewards. The same counterexample holds since we assumed $s_0$ and $s_1$ have the same reward. $\qquad\square$

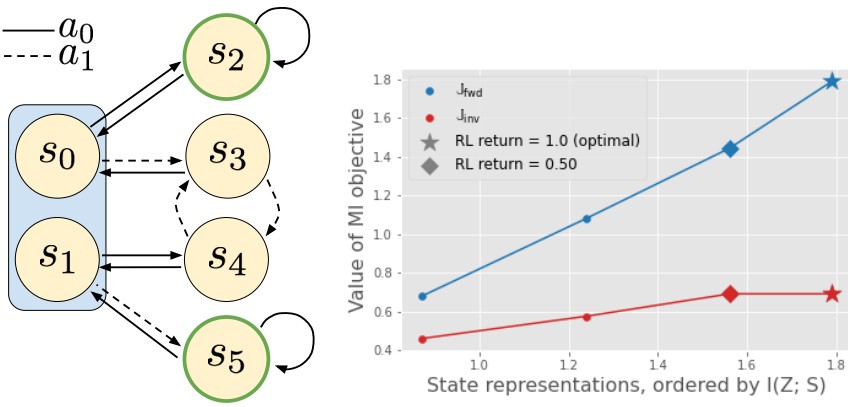

Figure 3: (left) In this MDP, a representation that aliases the states $s_0$ and $s_1$ into a single state maximizes $\mathbb{J}_{inv}$, yet is not sufficient to represent the optimal policy, which must distinguish between $s_0$ and $s_1$ in order to take a different action (towards the high-reward states outlined in green). (right) Values of $\mathbb{J}_{inv}$ and $\mathbb{J}_{fwd}$ for a few selected state representations, ordered by increasing $I(Z; S)$. The representation that aliases $s_0$ and $s_1$ (plotted with a diamond) maximizes $\mathbb{J}_{inv}$, but is not sufficient to learn the optimal policy. Note that this counterexample holds also for $\mathbb{J}_{inv} + I(R; Z)$.

# 6 EXPERIMENTS

In this section, we present experiments studying MI-based representation learning with image observations, to analyze whether the conclusions of our theoretical analysis hold in practice. Our goal is not to show that any particular method is necessarily better or worse, but rather to illustrate that the sufficiency arguments that we presented translate into quantifiable performance differences in the deep RL setting.

## 6.1 EXPERIMENTAL SETUP

To separate representation learning from RL, we first optimize each representation learning objective on a dataset of offline data consisting of 50k transitions collected from a uniform random policy. We then freeze the weights of the state encoder learned in the first phase and train RL agents with the representation as state input. To clearly illustrate the characteristics of each objective, we use the simple pygame (Shinners, 2011) video game *catcher*, in which the agent controls a paddle that it can move back and forth to catch fruit that falls from the top of the screen (see Figure 4). A positive reward is given when the fruit is caught and a negative reward when the fruit is not caught. The episode terminates after one piece of fruit falls. We optimize $\mathbb{J}_{fwd}$ and $\mathbb{J}_{state}$ with noise contrastive estimation (Gutmann & Hyvärinen, 2010), and $\mathbb{J}_{inv}$ by training an inverse model via maximum likelihood. For the RL algorithm, we use the Soft Actor-Critic algorithm Haarnoja et al. (2018), modified slightly for the discrete action distribution. Please see Appendix 8.2 for full experimental details.

## 6.2 COMPUTATIONAL RESULTS

In principle, we expect that a representation learned with $\mathbb{J}_{inv}$ may not sufficient to solve the *catcher* game. Because the agent does not control the fruit, a representation maximizing $\mathbb{J}_{inv}$ might discard that information, thereby making it impossible to represent the optimal policy. We observe in Figure 5 (top left) that indeed representations trained to maximize $\mathbb{J}_{inv}$ result in RL agents that converge slower and to a lower asymptotic expected return. Further, attempting to learn a decoder from the learned representation to the position of the falling fruit incurs a high error (Figure 5, bottom left), indicating that the fruit is not precisely captured by the representation. We argue that this type of problem setting is not contrived, and is representative of many situations in realistic tasks. Consider, for instance, an autonomous vehicle that is stopped

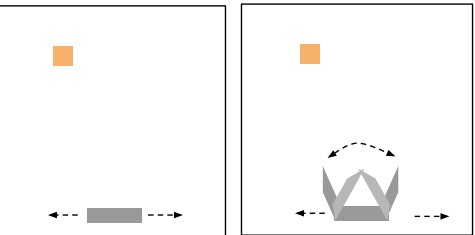

Figure 4: (left) Original *catcher* game in which the agent (grey paddle) moves left or right to catch fruit (yellow square) that falls from the top of the screen. (right) Variation *catcher-grip* in which the agent is instantiated as a gripper, and must open the gripper to catch fruit.

at a stoplight. Because the agent does not control the color of the stoplight, it may not be captured in the representation learned by $\mathbb{J}_{inv}$ and the resulting RL policy may choose to run the light.

In the second experiment, we consider a failure mode of $\mathbb{J}_{state}$. We augment the paddle with a gripper that the agent controls and must be open in order to properly catch the fruit. Since the change in the gripper is completely controlled by a single action, the current state contains no information about the state of the gripper in the future. Therefore, a representation maximizing $\mathbb{J}_{state}$ might alias states where the gripper is open with states where the gripper is closed. In our experiment, we see that the error in predicting the state of the gripper from the representation learned via $\mathbb{J}_{state}$ is chance (Figure 5, bottom right). This degrades the performance of an RL agent trained with this state representation since the best the agent can do is move under the fruit and randomly open or close the gripper (Figure 5, top right). In the driving example, suppose turning on the headlights incurs positive reward if it's raining but negative reward if it's sunny. The representation could fail to distinguish the state of the headlights, making it impossible to learn when to properly use the headlights.

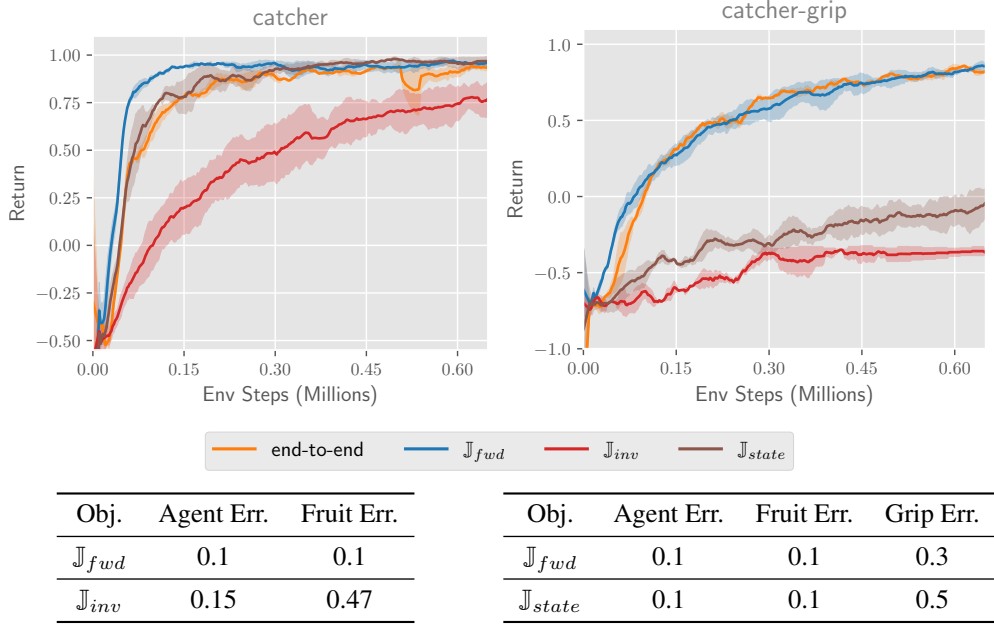

| Obj. | Agent Err. | Fruit Err. |
|------|-----------|-----------|
| $\mathbb{J}_{fwd}$ | 0.1 | 0.1 |
| $\mathbb{J}_{inv}$ | 0.15 | 0.47 |

| Obj. | Agent Err. | Fruit Err. | Grip Err. |
|------|-----------|-----------|-----------|
| $\mathbb{J}_{fwd}$ | 0.1 | 0.1 | 0.3 |
| $\mathbb{J}_{state}$ | 0.1 | 0.1 | 0.5 |

Figure 5: (top) Policy performance using learned representations as state inputs to RL, for the *catcher* and *catcher-grip* environments. (bottom) Error in predicting the positions of ground truth state elements from each learned representation. Representations maximizing $\mathbb{J}_{inv}$ need not represent the fruit, while representations maximizing $\mathbb{J}_{state}$ need not represent the gripper, leading these representations to perform poorly in *catcher* and *catcher-grip* respectively.

$\mathbb{J}_{fwd}$ produces useful representations in all cases, and is equally or more effective than learning representations purely from the RL objective alone (as in Figure 5). We experiment with more visual complexity by adding background distractors; these results are presented in Appendix 8.4. We find that in this setting representations learned with $\mathbb{J}_{fwd}$ to yield even larger gains over learning representations end-to-end via RL. We also analyze the learned representations by evaluating how well they predict the optimal $Q^*$ in Appendix 8.3.

## 7 DISCUSSION

In this work, we aimed to analyze mutual information representation learning objectives for control from a theoretical perspective. In contrast to much prior work that studies how these objectives can be effectively optimized given high-dimensional observations, we analyze which objectives are guaranteed to yield representations that are actually sufficient for learning the optimal policy. Surprisingly, we show that two common objectives yield representations that are theoretically insufficient, and provide a proof of sufficiency for a third. We validate our theoretical results with an empirical investigation on a simple video game environment, and show that the insufficiency of these objectives can degrade the performance of deep RL agents.

We view this investigation as a step forward in understanding the theoretical characteristics of representation learning techniques commonly used in deep RL. We see many exciting avenues for future work. First, identifying more restrictive MDP classes in which insufficient objectives are in fact sufficient, and relating these to realistic applications. Second, investigating if sample complexity bounds can be established in the case of a sufficient objective. Third, extending our analysis to the partially observed setting, which is more reflective of practical applications. We see these directions as fruitful in providing a deeper understanding of the learning dynamics of deep RL, and potentially yielding novel algorithms for provably accelerating RL with representation learning.

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

## 8 APPENDIX

### 8.1 SUFFICIENCY OF $\mathbb{J}_{fwd}$: PROOF OF PROPOSITION 1

We describe the proofs for the sufficiency results from Section 5 here. We begin by providing a set of lemmas, before proving the sufficiency of $\mathbb{J}_{fwd}$.

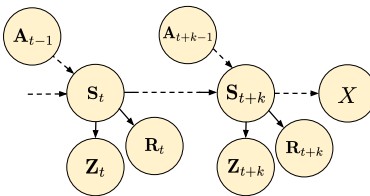

Figure 6: Graphical model for Lemma 1, depicting true states $S$, states in the representation $Z$, actions $A$, rewards $R$, and the variable $X$ (which we will interpret as the sum of future rewards in the proof of Proposition 1).

**Lemma 1.** *Let $X$ be a random variable dependent on $S_{t+k}$, with the conditional independence assumptions implied by the graphical model in Figure 6. (In the main proof of Proposition 1, we will let $X$ be the sum of rewards from time $t + k$ onwards.) If $I(Z_{t+k}; Z_t, A_t) = I(S_{t+k}; S_t, A_t)\forall k$, then $I(X; Z_t, A_t) = I(X; S_t, A_t)\forall k$.*

*Proof.* For proof by contradiction, assume there is some $\phi_{\mathcal{Z}}$ and some $r$ such that $I(X; Z_t, A_t) < I(X; S_t, A_t)$ and that $I(Z_{t+k}; Z_t, A_t) = I(S_{t+k}; S_t, A_t)$. Now we know that because $Z_t \rightarrow S_t \rightarrow S_{t+k} \rightarrow Z_{t+k}$ form a Markov chain, by the data processing inequality (DPI) $I(Z_{t+k}; Z_t, A_t) \leq I(S_{t+k}; Z_t, A_t) \leq I(S_{t+k}; S_t, A_t)$. We will proceed by showing that that $I(X; Z_t, A_t) < I(X; S_t, A_t) \implies I(S_{t+k}; Z_t, A_t) < I(S_{t+k}; S_t, A_t)$, which gives the needed contradiction.

Using chain rule, we can expand the following expression in two different ways.

$$I(X; Z_t, S_t, A_t) = I(X; Z_t|S_t, A_t) + I(X; S_t, A_t) = 0 + I(X; S_t, A_t) \qquad (5)$$

$$I(X; Z_t, S_t, A_t) = I(X; S_t|Z_t, A_t) + I(X; Z_t, A_t) \qquad (6)$$

Note that the first term in Equation 5 is zero by the conditional independence assumptions in Figure 6. Equating the expansions, we can see that to satisfy our assumption that $I(X; Z_t, A_t) < I(X; S_t, A_t)$, we must have that $I(X; S_t|Z_t, A_t) > 0$.

Now we follow a similar procedure to expand the following expression:

$$I(S_{t+k}; Z_t, S_t, A_t) = I(S_{t+k}; Z_t|S_t, A_t) + I(S_{t+k}; S_t, A_t) = 0 + I(S_{t+k}; S_t, A_t) \qquad (7)$$

$$I(S_{t+k}; Z_t, S_t, A_t) = I(S_{t+k}; S_t|Z_t, A_t) + I(S_{t+k}; Z_t, A_t) \qquad (8)$$

The first term in Equation 7 is zero by the conditional independence assumptions in Figure 6. Comparing the first term in Equation 8 with the first term in Equation 6, we see because $S_t \rightarrow S_{t+k} \rightarrow X$ form a Markov chain, by the DPI that $I(S_{t+k}; S_t|Z_t, A_t) \geq I(X; S_t|Z_t, A_t)$. Therefore we must have $I(S_{t+k}; S_t|Z_t, A_t) > 0$. Combining Equations 7 and 8:

$$I(S_{t+k}; S_t, A_t) = I(S_{t+k}; S_t|Z_t, A_t) + I(S_{t+k}; Z_t, A_t) \qquad (9)$$

Since $I(S_{t+k}; S_t|Z_t, A_t) > 0$, $I(S_{t+k}; Z_t, A_t) < I(S_{t+k}; S_t, A_t)$, which is exactly the contradiction we set out to show. $\qquad \square$

**Lemma 2.** *If $I(Y; Z) = I(Y; X)$ and $Y \perp Z|X$, then $\exists p(Z|X)$ s.t. $\forall x, p(Y|X = x) = \int p(Y|Z)p(Z|X = x)dz$.*

*Proof.* First note that the statement is not trivially true. Without any assumption regarding MI, we can write,

$$p(Y|X = x) = \int p(Y, Z|X = x)dz = \int p(Y|Z, X = x)p(Z|X = x)dz \tag{10}$$

Comparing this with the statement we'd like to prove, we can see that the key idea is to show that the MI equivalence implies that $p(Y|Z, X = x) = p(Y|Z)$. To begin, consider $I(Y; Z) = I(Y; X)$. We can re-write this equality using the entropy definition of MI.

$$H(Y) - H(Y|Z) = H(Y) - H(Y|X) \tag{11}$$

Note that the $H(Y)$ cancel and substituting the definition of entropy we have:

$$\mathbb{E}_{p(Y,Z)}[\log p(Y|Z)] = \mathbb{E}_{p(Y,X)}[\log p(Y|X)] \tag{12}$$

Note that on the right-hand side, we can use the Tower property to re-write the expectation as

$$\mathbb{E}_{p(Y,X)}[\log p(Y|X)] = \mathbb{E}_{p(Z)}\mathbb{E}_{p(Y,X|Z)}[\log p(Y|X)] = \mathbb{E}_{p(Y|X)p(X,Z)}[\log p(Y|X)] \tag{13}$$

Now we can use the Tower property again to re-write the expectation on both sides.

$$\begin{aligned}
\mathbb{E}_{p(X)}[\mathbb{E}_{p(Y,Z|X)}[\log p(Y|Z)]] &= \mathbb{E}_{p(X)}[\mathbb{E}_{p(Y|X)p(X,Z|X)}[\log p(Y|X)]] \\
\mathbb{E}_{p(X)}[\mathbb{E}_{p(Y|X)p(Z|X)}[\log p(Y|Z)]] &= \mathbb{E}_{p(X)}[\mathbb{E}_{p(Y|X)p(Z|X)}[\log p(Y|X)]] \\
\mathbb{E}_{p(X)}[\mathbb{E}_{p(Y|X)p(Z|X)}[\log p(Y|Z)]] &- \log p(Y|X)] = 0 \\
\mathbb{E}_{p(X)}[\mathbb{E}_{p(Y|X)}[\mathbb{E}_{p(Z|X)}[\log p(Y|Z)] &- \log p(Y|X)]] = 0
\end{aligned} \tag{14}$$

Log probabilities are always $\leq 0$, therefore for the sum to equal zero, each term must be zero.

$$\log p(Y|X) = \mathbb{E}_{p(Z|X)}[\log p(Y|Z)] \tag{15}$$

By Jensen's inequality,

$$\log p(Y|X) \leq \log \mathbb{E}_{p(Z|X)}[p(Y|Z)] \tag{16}$$

By the monotonicity of the logarithm:

$$p(Y|X) \leq \mathbb{E}_{p(Z|X)}[p(Y|Z)] \tag{17}$$

If there exists some $x$ and some $y$ such that $p(Y = y|X = x) < \mathbb{E}_{p(Z|X=x)}[p(Y = y|Z)]$, then there must be some other $y'$ for the same $x$ where $p(Y = y'|X = x) > \mathbb{E}_{p(Z|X=x)}[p(Y = y'|Z)]$ because $p(Y|X = x)$ must sum to 1.

$$p(Y|X) = \mathbb{E}_{p(Z|X)}[p(Y|Z)] = \int p(Y|Z)p(Z|X = x)dz = \int p(Y, Z|X = x)dz \tag{18}$$

Where the last equality follows by conditional independence of $Y$ and $Z$ given $X$. □

Given the lemmas stated above, we can then use them to prove the sufficiency of $\mathbb{J}_{fwd}$.

**Proposition 1.** *(Sufficiency of $\mathbb{J}_{fwd}$) Let $(\mathcal{S}, \mathcal{A}, \mathcal{T}, r)$ be an MDP with dynamics $p(S_{t+1}|S_t, A_t)$. Let the policy distribution $p(A|S)$ and steady-state state occupancy $p(S)$ have full support on the action and state alphabets $\mathcal{A}$ and $\mathcal{S}$ respectively. See Figure 6 for a graphical depiction of the conditional independence relationships between variables.*

*For a representation $\phi_{\mathcal{Z}}$ and set of reward functions $\mathcal{R}$, if $I(Z_{t+k}; Z_t, A_t)$ is maximized $\forall k > 0, t > 0$ then $\forall r \in \mathcal{R}$ and $\forall \mathbf{s}_1, \mathbf{s}_2 \in \mathcal{S}, \phi_{\mathcal{Z}}(\mathbf{s}_1) = \phi_{\mathcal{Z}}(\mathbf{s}_2) \implies \forall \mathbf{a}, Q_r^*(\mathbf{a}, \mathbf{s}_1) = Q_r^*(\mathbf{a}, \mathbf{s}_2)$.*

*Proof.* Note that $(Z_{t+k}; Z_t, A_t)$ is maximized if the representation $\phi_{\mathcal{Z}}$ is taken to be the identity. In other words $\max_\phi I(Z_{t+k}; Z_t, A_t) = I(S_{t+k}; S_t, A_t)$.

Define the random variable $\bar{R}_t$ to be the discounted return starting from state $\mathbf{s}_t$.

$$\bar{R}_t = \sum_{k=1}^{H-t} \gamma^k R_{t+k} \tag{19}$$

Plug in $\bar{R}_t$ for the random variable $X$ in Lemma 1:

$$I(Z_{t+k}; Z_t, A_t) = I(S_{t+k}; S_t, A_t) \qquad \Longrightarrow \qquad I(\bar{R}_{t+k}; Z_t, A_t) = I(\bar{R}_{t+k}; S_t, A_t) \quad (20)$$

Now let $X = [S_t, A_t]$, $Y = \bar{R}_t$, and $Z = Z_t$, and note that by the structure of the graphical model in Figure 6, $Y \perp Z | X$. Plugging into Lemma 2:

$$\mathbb{E}_{p(\mathbf{z}_t | S_t = \mathbf{s})} p(\bar{R}_t | Z_t, A_t) = p(\bar{R}_t | S_t = \mathbf{s}, A_t) \tag{21}$$

Now the $Q$-function given a reward function $r$ and a state-action pair $(\mathbf{s}, \mathbf{a})$ can be written as an expectation of this random variable $\bar{R}_t$, given $S_t = \mathbf{s}$ and $A = \mathbf{a}$. (Note that $p(\bar{R}_t | S_t = \mathbf{s}, A_t = \mathbf{a})$ can be calculated from the dynamics, policy, and reward distributions.)

$$Q_r(\mathbf{s}, \mathbf{a}) = \mathbb{E}_{p(\bar{R}_t | S_t = \mathbf{s}, A_t = \mathbf{a})}[\bar{R}_t] \tag{22}$$

Since $\phi_{\mathcal{Z}}(\mathbf{s}_1) = \phi_{\mathcal{Z}}(\mathbf{s}_2)$, $p(\mathbf{z}_t | S_t = \mathbf{s}_1) = p(\mathbf{z}_t | S_t = \mathbf{s}_2)$. Therefore by Equation 21, $p(\bar{R}_t | S_t = \mathbf{s}_1, A_t) = p(\bar{R}_t | S_t = \mathbf{s}_2, A_t)$. Plugging this result into Equation 22, $Q_r(\mathbf{a}, \mathbf{s}_1) = Q_r(\mathbf{a}, \mathbf{s}_2)$. Because this reasoning holds for all $Q$-functions [2], it also holds for the optimal $Q$, therefore $Q_r^*(\mathbf{a}, \mathbf{s}_1) = Q_r^*(\mathbf{a}, \mathbf{s}_2)$.

$\square$

## 8.2 EXPERIMENTAL DETAILS

### 8.2.1 DIDACTIC EXPERIMENTS

The didactic examples are computed as follows. Given the list of states in the MDP, we compute the possible representations, restricting our focus to representations that group states into "blocks." We do this because there are infinite stochastic representations and the MI expressions we consider are not convex in the parameters of $p(Z|S)$, making searching over these representations difficult. Given each state representation, we compute the value of the MI objective as well as the optimal value function using exact value iteration. In these examples, we assume that the policy distribution is uniform, and that the environment dynamics are deterministic. Since we consider the infinite horizon setting, we use the steady-state state occupancy in our calculations.

### 8.2.2 DEEP RL EXPERIMENTS

The deep RL experiments with the catcher game are conducted as follows. First, we use a uniform random policy to collect 50k transitions in the environment. In this simple environment, the uniform random policy suffices to visit all states (the random agent is capable of accidentally catching the fruit, for example). Next, each representation learning objective is maximized on this dataset. For all objectives, the images are pre-processed in the same manner (resized to 64x64 pixels and normalized) and embedded with a convolutional network. The convolutional encoder consists of five convolutional layers with ReLU activations and produces a latent vector with dimension 256. We use the latent vector to estimate each mutual information objective, as described below.

**Inverse information**: We interpret the latent embeddings of the images $S_t$ and $S_{t+1}$ as the parameters of Gaussian distributions $p(Z|S_t)$ and $p(Z|S_{t+1})$. We obtain a single sample from each of these two distributions, concatenate them and pass them through a single linear layer to predict the action. The objective we maximize is the cross-entropy of the predicted actions with the true actions, as in Agrawal et al. 2016 and Shelhamer et al. 2016. To prevent recovering the trivial solution of preserving all the information in the image, we add an information bottleneck to the image embeddings. We tune the Lagrange multiplier on this bottleneck such that the action prediction loss remains the same value as when trained without the bottleneck. This approximates the objective $\min_\phi I(Z; S) s.t. I_{inv} = \max I_{inv}$. To use the learned encoder for RL, we embed the image from the current timestep and take the mean of the predicted distribution as the state for the RL agent.

---

[2] Note this result is stronger than what we needed: it means that representations that maximize $\mathbb{J}_{fwd}$ are guaranteed to be able to represent even sub-optimal $Q$-functions. This makes sense in light of the fact that the proof holds for all reward functions - the sub-optimal $Q$ under one reward is the optimal $Q$ under another.

**State-only information**: We follow the Noise Contrastive Estimation (NCE) approach presented in CPC (Oord et al. 2018). Denoting $Z_t$ and $Zt+1$ as the latent embedding vectors from the convolutional encoders, we use a log-bilinear model as in CPC to compute the score: $f(Z_t, Z_{t+1}) = \exp(Z_t^T W Z_{t+1})$ for the cross-entropy loss. We also experimented with an information bottleneck as described above, but found that it wasn't needed to obtain insufficient representations. To use the learned encoder for RL, we embed the image from the current timestep and use this latent vector as the state for the RL agent.

**Forward information**: We follow the same NCE strategy as for state-only information, with the difference that we concatenate the action to $Z_t$ before computing the contrastive loss.

We then freeze the state encoder learned via MI-maximization and use the representation as the state input for RL. The RL agent is trained using the Soft Actor-Critic algorithm Haarnoja et al. (2018), modified slightly for the discrete action distribution (the Q-function outputs Q-values for all actions rather than taking action as input, the policy outputs the action distribution rather than parameters of a distribution, and we can directly compute the expectation in the critic loss rather than sampling). The policy and critic networks consist of two hidden linear layers of 200 units each. We use ReLU activations.

### 8.3 ANALYSIS: PREDICTING $Q^*$ FROM THE REPRESENTATION

In Section 6, we evaluated the learned representations by running a temporal difference RL algorithm with the representation as the state input. In this section, instead of using the bootstrap to learn the $Q$-function, we instead regress the $Q$-function to the optimal $Q^*$. To do this, we first compute the (roughly) optimal $Q^*$ by running RL with ground truth game state as input and taking the learned $Q$ as $Q^*$. Then, we instantiate a new RL agent and train it with the learned image representation as input, regressing the $Q$-function directly onto the values of $Q^*$. We evaluate the policy derived from this new $Q$-function, and plot the results for both the *catcher* and *catcher-grip* environments in Figure 7. We find that similar to the result achieved using the bootstrap, the policy performs poorly when using representations learned by insufficient objectives ($\mathbb{J}_{inv}$ in *catcher* and $\mathbb{J}_{state}$ in *catcher-grip*). Interestingly, we find that the error between the learned $Q$-values and the $Q^*$-values is roughly the same for sufficient and insufficient representations. We hypothesize that this discrepancy between $Q$-value error and policy performance is due to the fact that small differences in $Q$-values on a small set of states can result in significant behavior differences in the policy.

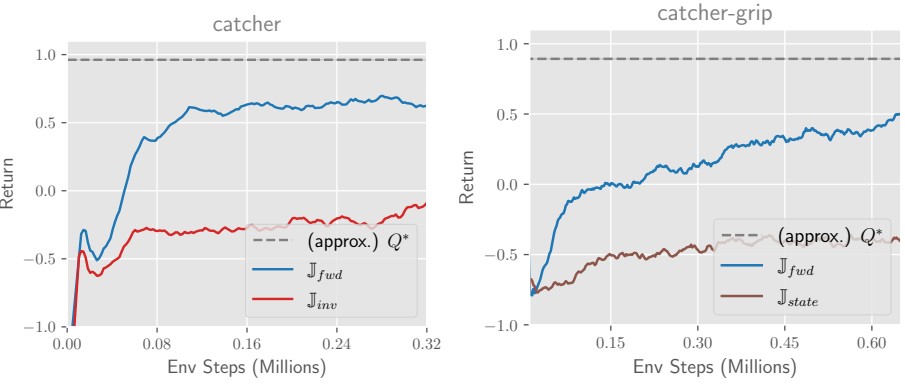

Figure 7: Performance of policies obtained from a $Q$-function trained to predict $Q^*$, given state representations learned by each MI objective, in the (left) *catcher* environment and (right) *catcher-grip* environment. Insufficient objectives $\mathbb{J}_{inv}$ and $\mathbb{J}_{state}$ respectively perform worse than sufficient objective $\mathbb{J}_{fwd}$.

### 8.4 DEEP RL EXPERIMENTS WITH BACKGROUND DISTRACTORS

In this section we repeat the experiments from Section 6 with added visual complexity in the form of background distractors. We randomly generate images of 10 circles of different colors and replace

the black background of the game with these images. Examples of the agent's observations are shown in Figure 8.

We plot the results for both the *catcher* and *catcher-grip* games with distractors in Figure 9. As in Section 6, we show both the result of performing RL with the frozen representation as input (top), as well as the error of decoding true state elements from the representation (bottom). In both environments, end-to-end RL from images performs poorly, demonstrating the need for representation learning to aid in solving the task. As predicted by the theory, the representation learned by $\mathbb{J}_{inv}$ fails in both games, and the representation learned by $\mathbb{J}_{state}$ fails in the *catcher-grip* game. We find that the difference in performance between sufficient and insufficient

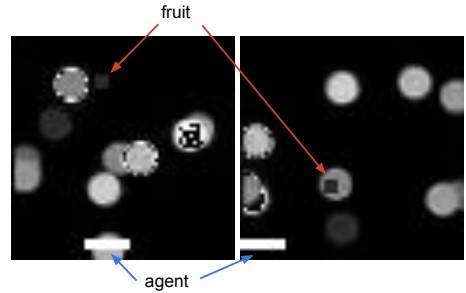

Figure 8: Example 64x64 pixel observations with background distractors.

objectives is even more pronounced in this setting than in the plain background setting.

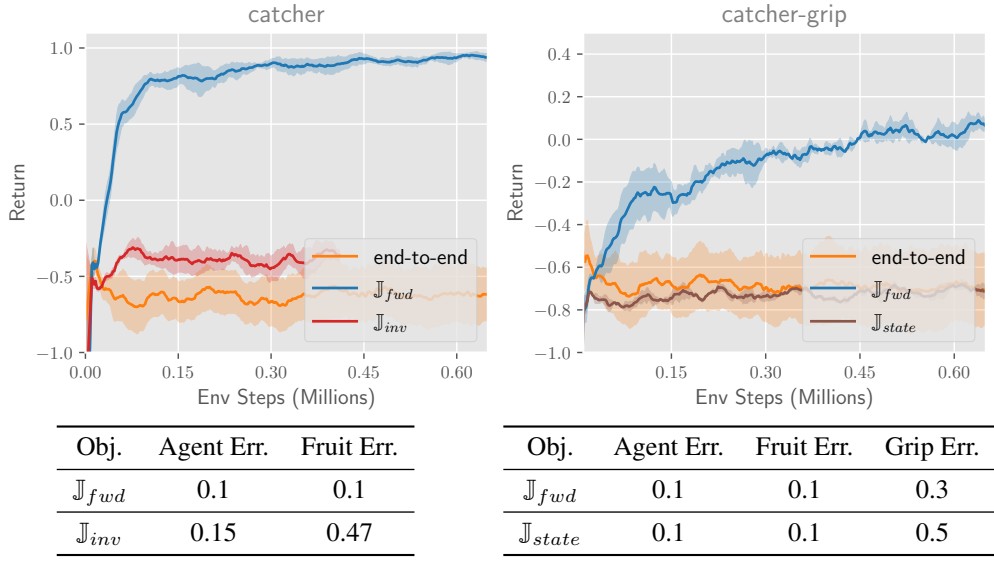

| Obj. | Agent Err. | Fruit Err. |
|---|---|---|
| $\mathbb{J}_{fwd}$ | 0.1 | 0.1 |
| $\mathbb{J}_{inv}$ | 0.15 | 0.47 |

| Obj. | Agent Err. | Fruit Err. | Grip Err. |
|---|---|---|---|
| $\mathbb{J}_{fwd}$ | 0.1 | 0.1 | 0.3 |
| $\mathbb{J}_{state}$ | 0.1 | 0.1 | 0.5 |

Figure 9: (top) Policy performance using learned representations as state inputs to RL, for the *catcher* and *catcher-grip* environments with background distractors. (bottom) Error in predicting the positions of ground truth state elements from each learned representation. Representations maximizing $\mathbb{J}_{inv}$ need not represent the fruit, while representations maximizing $\mathbb{J}_{state}$ need not represent the gripper, leading these representations to perform poorly in *catcher* and *catcher-grip* respectively.

