# OpenReview forum: "Which Mutual-Information Representation Learning Objectives are Sufficient for Control?"
_ICLR.cc/2021/Conference — Reject_

### Official Review · AnonReviewer4 · 2020-10-27
**Interesting perspective, but the state-only argument seems problematic.**

**Rating:** 5
**Confidence:** 4

**Review:**

**Update**
I have briefly checked the updated paper and the corresponding Proposition 1. While I currently do not find any issues with the counterexample, there are still many tiny issues in the proofs that prevent me from recommending acceptance.

- Prop 1 statement, $Q(a, s)$ appears again. It does not type check and not fixed in Def. 3.
- Lemma 2 statement, the last quantity, what is small $z$ in the integral? I also don't see why the $\exists$ symbol is there, isn't $Z$ as a random variable already defined, so $p(Z | X)$ is simply the conditional distribution?
- The step around (21) and (22) are very unclear. I can see what you are trying to do here, but I think it should be laid out step by step.

**Overview**
The paper discuss on a theoretical level what mutual information related quantities are well-suited for representation learning in the context of reinforcement learning. The paper argues that for reinforcement learning, forward information is the only one that is well suited for learning representations by a certain sufficiency definition, and state-only transition information and inverse information do not satisfy the sufficiency property; thus only the forward information is *a well-suited principle* for representation learning in reinforcement learning. Empirical results on simple offline data suggest validity of the theory.

**Strengths**
I think this paper is an interesting read, and that it is justified to consider a very simplified view over this setting (mutual information and be well approximated from samples, exploration is a non-issue, etc.). It can be used to discuss which MI based objective is adequate for RL in RL.

**Weaknesses**
I think the negative argument about state-only transition information is a bit flawed. The problem that I see here are two fold.
- One, arguments about the action distribution seems missing here, which should affect the representation distribution (also, it seems that the action distribution is not restricted to depend on $Z$, although the optimal policy clearly does?)
- Two, the proof does not discuss timesteps beyond $t=0$.

More concretely, if we assume Figure 2 (left), then $s_0$ gives no information about $s_1$ or $s_2$, so the representation learning has the freedom to compress $s_1$ and $s_2$ together. But if we consider the next timestep, then the compressed representation might not be optimal. Suppose
- At time = 1, I have 0.8 probability on $s_1$, 0.2 probability on $s_2$.
- At time = 2, my action distribution ensures that I have 0.5 probability of landing on either $s_1$ and $s_2$ when I start from either $s_1$ or $s_2$ (the graph for a Figure 2 has a bug for s1 and s2 actions)

Then if I use compressed representation, state-only information is zero (because I compressed the representation to 1 state anyways), but if I used original representation, then $H(S_2 | S_1) = 0.5 \log 2$, and $H(S_1)$ is smaller because it is Bernoulli(0.2). Therefore, the counterexample does not show that state-only information is not adequate. I am inclined to believe that the claim is true with added assumptions over the action distribution, but the proof presented here is not correct.

Minor comments:
- Definition 3, Q(a, s) does not type check.
- In Lemma 1, what is equation 8?
- Proof of lemma 2 seems problematic: why is $H(A+B) - H(A | Y) - H(B | Y) = I(Y; A+B)$ true? If A and B are near opposites of each other, then $A+B \approx 0$, and $H(A + B | Y) \approx \infty > H(A | Y) + H(B | Y)$. I think you need additional independence assumptions.

---

> ### Author Response · Authors · 2020-11-21
> **Response to review; updated counterexample for state-only MI**
>
> Thank you for your detailed review!
>
> *>it seems that the action distribution is not restricted to depend on Z, although the optimal policy clearly does*
>
> Could you clarify what you mean here? We’re not sure we understand the question.
>
> *>State-only MI counterexample*
>
> Note that in the paper we consider the infinite horizon case, and therefore compute the MI with the state occupancy distribution which does not depend on the time step. However, you are correct that there was a bug in this counterexample. We have updated the paper with a working counterexample, and here we detail the calculation performed to compute the MI for that example.
> First we calculate $I(S_{t+k}; S_t)$. Let $k=1$ (this is sufficient for a counterexample since maximizing the objective for all $k>1$ will include $k=1$).
>
> $H(S) = 4 \frac{1}{4} \log(4) = 2 \log(2)$.
> $H(S_{t+1} | S_t) = 8 \frac{1}{2} \frac{1}{4} \log(2) = \log(2)$.
> Therefore $I(S_{t+1}; S_t) = 2 \log(2) - \log(2) = \log(2)$.
>
> Now consider the representation $Z$ that aliases $s_0$ and $s_3$.
> $H(Z) = \frac{1}{2} \log(2) + \frac{1}{2} \log(4) = \frac{3}{2} \log(2)$
> $H(Z_{t+1}| Z_t) = 2 \frac{1}{2} \frac{1}{2} \log(2) = \frac{1}{2} \log(2)$
> Therefore $I(Z_{t+1}; Z_t) = \frac{3}{2} \log(2) - \frac{1}{2} \log(2) = \log(2)$.
>
> The values of the MI objectives are the same, but the optimal policy cannot be recovered in the representation space. Suppose state $s_2$ has a reward of 1 while all other states have reward 0. Without distinguishing between $s_0$ and $s_3$ the policy cannot choose the action that leads to $s_2$.
>
> *>Definition 3, Q(a, s) does not type check.*
>
> Thanks for pointing out this typo, it has been fixed in the updated paper. It should read $\phi(s_1) = \phi(s_2)$, lowercase (particular state value) not uppercase (random variable for state).
>
> *>In Lemma 1, what is equation 8?*
>
> Thanks for pointing out this missing reference, we have fixed it. We meant to refer to the expansion of $I(Z_t, S_t, A_t; R_{t+1})$.
>
> *>Proof of lemma 2 seems problematic*
>
> Thank you for pointing out this bug in the proof of Proposition 1. You are correct that as written, Lemma 2 is false. We have updated the proof of Proposition 1, and would be happy to know if you spot any other issues!

---

> > ### Comment · AnonReviewer4 · 2020-11-22
> > **Thank you for the update**
> >
> > > it seems that the action distribution is not restricted to depend on Z, although the optimal policy clearly does
> >
> > I think the key issue I had in mind was, how would different action distributions affect the distribution of state action pairs, and consequently the mutual information value that is evaluated. Because you care about $\pi^\star$ and $Q^\star$-sufficiency, then it seems that 1) the underlying action distribution depends on the optimal policy; 2) if some set of transitions are not very relevant to $\pi^\star$, then maybe it is okay to not encode them into $z$.
> >
> > Then, how can we find the optimal policy? I suppose that the key for using representations is that learning the optimal policy would be easier under the representations, but this creates a chicken-and-egg problem. Without optimal policy, you are not supposed to have the state action distribution to evaluate for mutual information. But if we had the optimal policy, why would we need $z$ in the first place? This problem (state action distribution depends on policy) also relates to the counterexample in Figure 2, where I think you assumed the policy chooses each action with 0.5 probability (though I think the calculations are correct). Is this a fair and reasonable assumption?

---

> > > ### Author Response · Authors · 2020-11-23
> > > **Effect of state-action distribution on MI values and representations learned**
> > >
> > > It’s true that different policy distributions produce different values of the MI objective. Fortunately, the only assumption on the state-action distribution that Proposition 1 requires is that it has full support. In other words, while the value of $I(S_{t+k}; S_t, A_t)$ varies with the choice of $p(S, A)$, as long as the chosen $p(S, A)$ has full support, any representation maximizing $I(Z_{t+k}; Z_t, A_t)$ will be sufficient to learn and represent the optimal policy. There is no need to know the optimal policy or Q. The reason that full support of $p(S, A)$ is required is because to *learn* the optimal policy, and not just represent it, the representation must be able to represent states that lie outside the support of the optimal policy distribution. Therefore we can’t discard information solely based on its relevance to $\pi^*$ as suggested in your point (2).
> > >
> > > For the counterexamples in Figures 2 and 3, it suffices to show that there exists a policy distribution with full support for which the $\mathcal{J}_{state}$ objective is not sufficient. We use the uniform policy distribution in our examples for simplicity, which is arguably also a somewhat realistic choice of exploration policy. Note that this counterexample remains a counterexample with some other choices of policy distributions, for example $p(A | S) = \[0.75, 0.25\]$.
> > >
> > > Please let us know if you still have concerns!

---

### Official Review · AnonReviewer1 · 2020-10-27
**Interesting research question but limited empirical evaluation**

**Rating:** 5
**Confidence:** 3

**Review:**

Summary

This paper studies 3 mutual information (MI) optimization objectives for learning a latent representation Z in a sequential decision-making task of the MDP style:

forward information: MI between Z at time t+k, and Z at time t concatenated with the action at time t,

state-only transition information: MI between Z at time t+k, and Z at time t,

inverse information: MI between the action at time t, and Z at time t+k, conditioned on Z at time t.

All of the 3 objectives above are theoretically analysed with the result that only forward information can lead to a sufficient representation for reinforcement learning (based on the concept of Q*-sufficiency). This is experimentally confirmed in 2 simple games: catcher and gripper. In these 2 games, a latent representation based on forward information for reinforcement learning will eventually recover an end-to-end trained agent, while the other latent representations won't.

Quality and Details

The motivation of the paper is interesting and a systematic approach is taken to answer the question which representation learning objective is best, starting from a theoretical analysis and ending in an empirical study. However, I am a bit concerned about the experimental side of the paper which is a bit scarce.

Clarity

The paper is clearly written and easy to follow.

Originality and Significance

The research question is interesting and original, but the experiments only deal with 2 simple environments.

Pros

The paper has a good motivation and a systematic approach to answer the raised question.

Cons

The empirical evaluation is limited.

Minor

I have the feeling that the experiments in Figure 5 and 6 are missing baselines? In Figure 5, I can't see a state-only ablation, while in Figure 6, I can't see an inverse-information ablation?

Maybe I skipped it, but what was k in the experiments?

---

> ### Author Response · Authors · 2020-11-14
> **Clarifications about Fig. 5 and 6.**
>
> Thank you for your review. We will add further experiments (see the group response).
>
> *>I have the feeling that the experiments in Figure 5 and 6 are missing baselines?*
>
> In Fig. 5 and 6, the curves labeled as 'end-to-end' are the baseline of running RL without any prior representation learning, and the other curves show the performance of the same RL algorithm training on the state representations studied in our paper - each learned with the objective indicated in the subindex. We will further clarify this on the caption and text. Is there still anything the reviewer is missing?
>
> *>In Figure 5, I can't see a state-only ablation, while in Figure 6, I can't see an inverse-information ablation?*
>
> To reduce clutter, we didn’t plot the result for state-only information representation on the environment demonstrating the insufficiency of inverse information and vice-versa. We will add plots showing all the algorithms in all environments in the appendix for completness.
>
> *>Maybe I skipped it, but what was k in the experiments?*
>
> We used k=1 in the experiments.

---

> > ### Comment · AnonReviewer1 · 2020-11-17
> > **Re: Clarifications about Fig. 5 and 6.**
> >
> > Thanks for the clarification. I do not think that the plots would look cluttered if the mentioned baselines were added.

---

> > > ### Author Response · Authors · 2020-11-25
> > > **Methods have been added to plot**
> > >
> > > We have updated the plots with the the requested methods. Please see the group response for all the updates!

---

### Official Review · AnonReviewer2 · 2020-10-28
**Official Blind Review #2**

**Rating:** 5
**Confidence:** 3

**Review:**

Summary of the work: This work studies which mutual-information representation learning objectives (1. forward information, 2. state-only transition information, 3. inverse information) are sufficient for control in terms of representing the optimal policy, in the context of reinforcement learning (RL). As a result, they find a representation that maximizes 1 is sufficient for optimal control under any reward function, but 2 and 3 fails to provide that guarantee in some MDP cases. They provide both proof and interesting counter examples to justify the findings. Besides, they conduct some empirical studies on a video game (i.e. Catcher) and show that the sufficiency of a representation can have a substantial impact on the performance of an RL agent that uses that representation.

l like the idea of trying to understand the recent popular mutual information objectives in RL. To the best of my knowledge, Q^*-sufficiency analysis for mutual information objectives is novel. The counterexamples in sufficiency analysis are interesting. The paper is well written.

However, l still have the following concerns:

Empirically mutual information objectives often play the role of auxiliary losses to improve the sample efficiency of RL. Thus, it is very useful for us to theoretically understand which objective is better in terms of improving sample efficiency. The property of sufficiency (i.e. the ability to represent the optimal policy) is important. However, only this property may not strong enough, because there may exist a trade-off between minimizing information loss and maximizing state space reduction (Li et al. 2006). Can we add one more perspective, such as whose representation is finer (like Definition 2 in Li et al. 2006) to better understand these MI objectives, if possible? Intuitively, a coarser representation results in a larger reduction in the state space, which in turn translates into the efficiency of solving the problem.

Regarding the experiment results, the authors give some intuitive descriptions to show that state-only transition objective and inverse objective may be insufficient, but forward objective works in the catcher game. To enhance its solidness, l strongly suggest that we may conduct an experiment on the predictability of optimal Q-function by the representations trained by various MI objectives. For example, we can spend a long time to train a good enough policy and treat it as an optimal policy.


Regarding the modeling of representation, the representation is modeled as a random variable in this paper for analysis. However, in practice, the representation is always built upon neural networks and thus deterministic. Therefore, l am a little curious if the derived conclusion will still hold for deterministic cases.

Regarding the proof of Proposition 4, l am sorry that l do not fully understand the derivation from Q(s,a) to Q^* (s,a). Can the authors provide more details on that?

From the proof of proposition 4 in appendix,  I(Z_t,A_t;Z_(t+k)) seems to be maximized for ∀k>0,t>0. l suggest we make that clearer in the definition of those MI objectives (e.g. Equation 2), since some practical algorithms are based on the fixed k, not all k.

---

> ### Author Response · Authors · 2020-11-14
> **Finer mappings, stochasticity and Proposition 4**
>
> Thank you for your comments, we reply below to what we identify as the main concerns, let us know if there is anything that is not considered properly addressed.
>
> *>Can we add one more perspective, such as whose representation is finer (like Definition 2 in Li et al. 2006) to better understand these MI objectives, if possible?*
>
> We agree that which representation is finer (related to minimality) is an important criterion, as long as the set of representations being compared are all sufficient. Among sufficient representations, the most minimal representation would be preferred since it discards the most irrelevant information. The point we would like to make is that before we consider minimality or other desiderata, we should first ensure sufficiency, and that surprisingly we show that some common objectives fail this basic criterion - only one of the three common objectives studied in the paper is indeed sufficient!
>
> *>l strongly suggest that we may conduct an experiment on the predictability of optimal Q-function by the representations trained by various MI objectives.*
>
> Thank you for this suggestion, we agree it would strengthen the result. We will add this to the paper.
>
> *>Regarding the modeling of representation, the representation is modeled as a random variable in this paper for analysis. However, in practice, the representation is always built upon neural networks and thus deterministic.*
>
> While the neural network is deterministic, we can use it to parameterize a probability distribution over latent variables $Z \sim p_{\phi(s)}(Z|S=s)$. So while the resulting Z is stochastic, the mapping from state to the latent distribution parameters $\phi(s)$ is indeed deterministic.
>
> *>Regarding the proof of Proposition 4, l am sorry that l do not fully understand the derivation from* $Q(s,a)$ to $Q^*(s,a)$ . *Can the authors provide more details on that?*
>
> The last equation in the proof of Proposition 4 is the standard definition of the $Q$-function for any policy, and thanks to the equation derived just before we have that if the latent distribution $p(z_t | S_t=s, A)$ obtained by maximizing $J_{fwd}$ is the same for two different states $s_1$ and $s_2$ then the distribution of discounted future returns must be the same, and therefore their $Q$s are the same. Given that this holds true for all policies, it also holds true for the optimal policy - hence proving the proposition by establishing that $Q^*(s_1, a) = Q^*(s_2, a)$.
> We will number the equations and re-write the expectations such that the connection between the steps is more clear.
>
> *>From the proof of proposition 4 in appendix, $I(Z_t,A_t;Z_{t+k})$ seems to be maximized for ∀k>0,t>0. l suggest we make that clearer in the definition of those MI objectives (e.g. Equation 2)*
>
> Thanks for the suggestion, we will make sure to clarify the definition.

---

> > ### Author Response · Authors · 2020-11-22
> > **Any other concerns?**
> >
> > Dear reviewer,
> >
> > Do you have any other concerns we could address?
> > To respond to your initial review, we have clarified the proof of Proposition 1, and are working on the experiment predicting Q* (we will update the paper with it in the next couple days).

---

> > ### Comment · AnonReviewer2 · 2020-11-24
> > **Thank you for your response**
> >
> > Thank you for the clear and detailed response. However, l still have concerns that it might be not impactful to only analyze the sufficiency of representations, although necessary. Additionally, Rohin Shah and R4 have raised some concerns about the correctness of the proofs. Thus, l tend to stick with my original rating. I will keep track of the discussions and your responses to other reviewers.

---

> > > ### Author Response · Authors · 2020-11-25
> > > **Thanks for the response; clarification of status of proof of Prop 1**
> > >
> > > We have updated the proof of Proposition 1 to correct the mistake in the previous Lemma 2 noted by R4. The counterexample proposed by Rohin Shah is not correct. The $Q^*$ prediction experiments have been added to the paper (please see latest group response for all updates).

---

### Official Review · AnonReviewer3 · 2020-11-01
**Conditions in which mutual information objectives are sufficient for reinforcement learning**

**Rating:** 7
**Confidence:** 3

**Review:**

This paper studies which commonly-used mutual information objectives for learning state representations are sufficient for reinforcement learning. In particular, they provide counterexamples to show that state-only and inverse MI objectives are not Q*-sufficient, while proving that forward MI is Q*-sufficient. They validate their findings empirically with experiments in a simple RL domain.

There has been a lot of work recently in reinforcement learning that uses mutual information objectives resulting in performance gains, so it’s very fascinating to see a finding that these objectives may be theoretically insufficient, despite their empirical success. The counterexamples shown are simple and the authors do a good job of explaining the intuition. I think this paper will be of great interest to the ICLR community.

One question: while J_state and J_inv are not sufficient, are there conditions in which they can be? If these conditions are limited to just the reward, could this somehow give insight on how to design better reward functions?

Minor: there’s a reference in the last paragraph on page 6 to Figure 5 which I think should be to Figure 4.

---

> ### Author Response · Authors · 2020-11-25
> **Thanks for your review; speculation about sufficiency of J_state and J_fwd**
>
> Thanks for your comments!
>
> Regarding conditions for the sufficiency of $J_{state}$ and $J_{inv}$, we agree this is a very interesting question. While far from a systematic analysis, we can identify certain types of MDPs in which the objectives appear to be sufficient. For example, when the dynamics and policy are deterministic, $J_{state}$ should be sufficient, because in this case $p(S_{t+k}; S_t) = p(S_{t+k}; S_t, A_t)$. For $J_{inv}$, we suspect the objective could be sufficient when the reward depends on an aspect of the state that the agent cannot control. In this case, whether the agent represents that aspect or not does not affect representing the optimal policy. We plan to explore this question further in future work.
>
> Thanks for pointing out the typo, we've fixed it.

---

### Official Review · AnonReviewer5 · 2020-11-04
**Interesting and useful direction, but perhaps limited in clarity and applicability**

**Rating:** 6
**Confidence:** 3

**Review:**

**Summary**: The paper discusses three mutual information (MI) objectives for representation learning in RL, referred to as forward, state, and inverse. The forward MI objective models latent dependencies given the action. The state MI objective models latent dependencies alone. And the inverse MI objective models dependencies between actions and future states (empowerment). The paper shows that of these three common objectives, only the forward objective is sufficient for learning the optimal policy / value function. This is demonstrated using simple examples and experiments on a simple game environment.

**Strong Points**: This paper attempts to provide an overarching view of the many previous works that have used MI objectives for representation learning in RL. Multiple approaches are compared under one generic formulation. This can be helpful in connecting disparate research areas, e.g., connecting contrastive representation learning with empowerment and other task-agnostic policy learning objectives. Papers that present more general interpretations, such as this one, help to build consensus in the research community and standardize techniques, which can be more useful that proposing modifications of existing techniques.

Along similar lines, to the best of my knowledge, this paper presents a somewhat novel idea: analyzing MI objectives in terms of whether they are sufficient for (optimally) performing downstream tasks. Although representation learning and policy/value learning are not always performed separately, many works do not theoretically interrogate whether their representation learning objective is sufficient to perform the task. Having a clearer theoretical grounding can be helpful for deciding which representation learning objectives are worth pursuing.

Overall, the paper is well written. Key mathematical concepts are defined clearly in sections 3 and 4. The descriptions and definitions are clear and concise.

Although the experiments are rather limited, they do help to isolate key aspects of the analysis. In particular, I found the regression of components of the environment (fruit error and gripper error) to be compelling and helpful.

**Weak Points**: I found the technical formulation somewhat unclear. The paper presents representations as stochastic mappings from states (s) to latent variables (z). Sufficiency is defined as having the same optimal policies / value functions when the representations are the same. However, it is unclear, in practice, how such representations are intended to be used for value/policy learning. From the proof sketch in proposition 1, it seems as though the value is estimated by integrating over Z, however, this may not be feasible in practice, and we may need to use samples. Regarding this point, it is also unclear what is performed in the actual experiments of the paper.

It’s unclear whether analyzing the sufficiency of representations for downstream tasks is impactful for future work. Sufficiency (see definitions in section 3) describes whether the learned representation removes any task-relevant information. However, there are infinitely many representations that are sufficient. While sufficiency is certainly important for learning representations, it is only half of the consideration. In practice, one would want a *minimal* sufficient representation that is amenable for learning. Likewise, there are cases where representation learning and task-based learning occur together, in which case, task-based learning may be able to overcome the insufficiency of representation learning.

The paper is almost entirely lacking in experiment details. The descriptions for the examples in figures 2 and 3 could be improved in clarity, along with details on the estimation of each of the MI terms. Likewise, beyond basic descriptions of the tasks in the experiments section, very few details are present. While I understand that the focus of this paper is more theoretical in nature, such details are essential for reproducible results and ensuring technical rigor.

This paper may not be entirely representative of previous work. Mutual information objectives of differing types are applied in various contexts, in many cases trained during data collection. This paper explores a limited setting, in which these objectives are trained on data collected from a uniform policy and (I assume) the policy / value function is estimated only from the representation. While the authors claim that this enables a fair comparison between the objectives, it also somewhat limits the scope/impact of the paper, as it is less realistic. Further, experiments are performed on fairly limited, fully-observed environments. While I understand that these simple environments are meant to capture the essential differences in these objectives, focusing only on these relatively “toy” environments could limit the impact of this paper. For instance, in the two experiments in the experiments section, representation learning provides fairly marginal improvements in performance over just directly training end-to-end. Even just sticking with these environments, the paper could be improved with further analyses on the types of representations that are learned in each case.

**Accept / Reject**: Given the relative lack of details and the limited experimental investigation, I would lean slightly toward rejection. While I agree with the direction of this paper, I feel that these aspects would need to be improved for the paper to have substantial impact on the rest of the research community. With a more in-depth discussion of the experiment details and perhaps further analysis of what is and is not captured by each objective, this paper could reach the bar for acceptance. Experiments and analysis with previously published approaches would help to further improve the paper.

**Questions**:

Could you please provide more details on the training scheme in the experiments section. How is the representation used in practice for downstream tasks?

Where do other unsupervised representation learning schemes (e.g. VAEs and normalizing flows) fit within this framework?

For J_state, it is stated that previous values of Z can be ignored due to the Markov state. However, if the mapping from S to Z discards dynamics information, Z will not be Markovian. Is this formulation assumption still valid?

**Additional Feedback**:

Related Work:
Equations 2 and 4 are referenced far before they are defined. I generally avoid referencing equations this far in advance.
I would consider citing Mohamed & Rezende, 2015.

Representation Learning in RL:
Missing the discount factor in the preliminaries section.

Sufficiency Analysis:
Figure 2: J_inv —> J_state

Experiments:
Figure 6: should be J_state in the table.

---

> ### Author Response · Authors · 2020-11-14
> **Response to questions and concerns (1/2)**
>
> *>Stochastic mappings from states (s) to latent variables may require samples*
> This is a good observation! To be as general as possible, we allow for the representation parameters $\phi$ to parameterize a generic stochastic mapping $p_{\phi}(Z|S)$. So given a particular state, the representation produces a distribution over the latent space Z. This is required in many representation learning objectives, including all the ones based on Mutual Information formulations, VAEs, etc. The Q value for a particular state $s$ and action $a$ includes an expectation over $Z$, which can be approximated with samples.  In practice we observed in our experiments that using a single sample from the collapsed distribution gave the same resulting performance than using many samples, since the learned distributions are typically fairly narrow
>
> *>While sufficiency is certainly important for learning representations, it is only half of the consideration. In practice, one would want a minimal sufficient representation that is amenable for learning.*
> We agree that among sufficient representations, the most minimal representation would be preferred since it discards the most irrelevant information. The point we would like to make is that before we consider minimality or other desiderata, we should first ensure sufficiency, and that surprisingly some common objectives fail this basic criterion. It is possible to define minimality with respect to a certain representation objective by posing the optimization $\min I(Z; S) s.t. J_{MI} = \max J_{MI}$. We do this by maximizing the Lagrangian in the deep RL experiments in order to find the representations that maximize the MI objective but aren’t sufficient (see experimental details in the group response).
>
> *>there are cases where representation learning and task-based learning occur together, in which case, task-based learning may be able to overcome the insufficiency of representation learning.*
> From a theoretical standpoint, end-to-end RL is capable of learning exactly the representation it needs to solve the task. In practice, the dynamics of how exactly representation learning objectives improve RL optimization is an open question. While it’s possible that the unsupervised objective and RL objective work in tandem to encode different pieces of relevant information, that is not guaranteed in principle. In this paper, to disentangle these issues, we focused on what the unsupervised objectives can learn on their own, and we view this as a building block towards a theoretical understanding of common practices such as joint training.
>
> *>The paper is almost entirely lacking in experiment details.*
> Thanks for pointing this out - we will address this as fully detailed in the group response.
>
> *>This paper explores a limited setting, in which these objectives are trained on data collected from a uniform policy and (I assume) the policy / value function is estimated only from the representation.*
> It is correct that the policy/value function is estimated only from the representation and has no access to the underlying state, as commonly done when learning representations for RL.
> The proof of Proposition 4 relies on the assumption that the policy places some nonzero probability on every available action in every state - any exploration policy with this property (quite mild to satisfy and in fact desirable for any exploration policy) would still yield a sufficient representation when optimizing for I_fwd. In all computational experiments, we use the uniform policy to collect data for simplicity. As for the counterexamples showing that I_inv and I_state are not guaranteed to be sufficient, we could give many other non-uniform distributions that would also yield a non-sufficient representation.
>
> *> experiments are performed on fairly limited, fully-observed environments.*
> Most environments studied in RL are fully-observed, otherwise they would not be MDPs and the theoretical study would be considerably more complex. Extending to the partially observed case is an interesting avenue for future work.
>
> *>the paper could be improved with further analyses on the types of representations that are learned*
> To analyze what information the representation contains, we attempt to decode the underlying low-dimensional state from the representation (see the tables in Figures 5, 6). Could you expand on what you mean by “type of representation” and some potential analyses that might be appropriate?
>
> *>Experiments and analysis with previously published approaches would help to further improve the paper.*
> All the objectives we experiment with in the paper are previously published objectives (see experimental details). Are there other published approaches that you think would be relevant for comparison?

---

> > ### Author Response · Authors · 2020-11-14
> > **Response to questions and concerns (2/2)**
> >
> > *>Where do other unsupervised representation learning schemes (e.g. VAEs and normalizing flows) fit within this framework?*
> > We agree that this is a very interesting question. These generative models are based on a different graphical model from the one we assume in this paper (see Fig. 1) and hence we would need different tools to analyze them. However, we can still ask the question, what information must the representation contain? Prior work has shown that maximizing the ELBO alone cannot control the content of the learned representation (Huszar et al. 2017, Phuong et al. 2018), and Alemi et al. 2018 show that for the same ELBO value, the representation can contain no information about the input at all, all the information, or somewhere in between. Therefore we conjecture that the zero-distortion maximizer of the ELBO would be sufficient, while other solutions would not necessarily be. This is beyond the scope of our paper, but we will add it to the future directions section.
> >
> > *>For J_state, it is stated that previous values of Z can be ignored due to the Markov state. However, if the mapping from S to Z discards dynamics information, Z will not be Markovian. Is this formulation assumption still valid?*
> > Would you be able to point us towards where in the paper this statement is made? We’re not sure we understand the question in context. We hope we can address the concerns accordingly.
> >
> > Thank you for pointing out typos and the missing citation. We will correct these in an updated version of the paper.

---

> > > ### Author Response · Authors · 2020-11-22
> > > **Any other concerns?**
> > >
> > > Dear reviewer,
> > >
> > > Do you have any other concerns we could address?
> > > In response to your initial review, we have updated the paper with experimental details, and we are working on the additional experiments with background distractors to increase visual complexity (we will update the paper with the results of these experiments in the next couple days).

---

> > > ### Comment · AnonReviewer5 · 2020-11-22
> > > **Thank you for your response**
> > >
> > > Thank you for the detailed response to my review and addressing my questions and concerns. As stated above, one of my concerns was a lack of experiment details, so I am glad to see that these details have been added to the paper. I have increased my score to a 6 reflect this improvement.
> > >
> > > However, I still have concerns that this paper may be too far removed from typical experimental settings to have a real impact on the field. The other reviewers and I agree that the premise of the paper is interesting and useful, but due to particular aspects of the investigation (data collection using the random policy, stage-wise representation and task training, fairly toy environments), it's not clear that this paper will impact the way people approach mutual information objectives. Additionally, other reviewers have raised concerns over the correctness of the proofs. I will wait to see further updates and the developments of discussions with other reviewers before increasing my score further.

---

> > > > ### Author Response · Authors · 2020-11-23
> > > > **Thank you for the update; response to remaining concerns**
> > > >
> > > > Thank you for your reply and the revised score. Below we briefly clarify the status of the proof of Proposition 1, and some of our choices in regards to experimental setup.
> > > >
> > > > *>other reviewers have raised concerns over the correctness of the proofs*
> > > >
> > > > We have updated the proof in response to the concerns from R4. To the best of our knowledge, the proof is now correct.
> > > >
> > > > *>too far removed from typical experimental settings to have a real impact on the field*
> > > >
> > > > We view the most important contribution of this paper as theoretical, which remains valid regardless of the complexity of the experiments. The performance of representation learning algorithms for RL in practice is a function of many factors: the difficulty of MI estimation in high-dimensional spaces, noise and bias in the data, and function approximation in RL, to name a few. We think our sufficiency results point to another factor to consider when evaluating and proposing new representation learning algorithms.
> > > >
> > > > *>data collection using the random policy*
> > > >
> > > > Note that Proposition 1 holds as long as $p(S, A)$ has full support, so many other distributions besides the uniform distribution could be used to collect data. As R4 notes, having to assume anything about the optimal policy, or requiring any prior knowledge on the environment might defeat the purpose of learning representations. Nevertheless, if the user happens to already have some way of biasing the exploration towards more interesting states this can be leveraged also.
> > > >
> > > > *>stage-wise representation and task training*
> > > >
> > > > We perform experiments stage-wise in order to study the characteristics of the representations learned by the MI objectives. If we optimized the MI objective and RL objective jointly, it would be impossible to analyze this without the effect of the RL objective. We’d also like to point out that stage-wise training is an approach used in practice for making use of unlabeled data to reduce reliance on costly reward supervision (Lange et al. 2012, Finn et al. 2016, Ghadirzadeh et al. 2017).
> > > >
> > > > **References**
> > > >
> > > > Finn, Chelsea and Tan, Xin Yu and Duan, Yan and Darrell, Trevor and Levine, Sergey and Abbeel, Pieter. *Deep spatial autoencoders for visuomotor learning.* 2016 IEEE International Conference on Robotics and Automation (ICRA)
> > > >
> > > > Lange, Sascha and Riedmiller, Martin and Voigtländer, Arne. *Autonomous reinforcement learning on raw visual input data in a real world application.* The 2012 international joint conference on neural networks (IJCNN)
> > > >
> > > > Ghadirzadeh, Ali and Maki, Atsuto and Kragic, Danica and Björkman, Mårten. *Deep predictive policy training using reinforcement learning.* 2017 IEEE/RSJ International Conference on Intelligent Robots and Systems (IROS)

---

### Author Response · Authors · 2020-11-14
**Experiment Details**

We’d like to thank all the reviewers for their insightful reviews! We’ll respond to each reviewer’s points in direct replies to each reviewer, but here we’d like to clarify experimental details and explain the further experiments we will conduct.

**Further experiments**
As suggested by R2, we will train a $Q^*$ predictor from the learned representation via supervised learning. This will help directly evaluate if $Q^*$ is representable, which will complement the existing RL and state regression experiments.

To evaluate if our results hold on a more visually complex domain, we will conduct experiments with background distractors where end-to-end RL performs poorly.

**Experimental Details**
Below we include a summary of experimental details, and we will update the paper with a detailed appendix.
The didactic examples are computed as follows.
Given the list of states in the MDP, we compute the possible representations, restricting our focus to representations that group states into “blocks.”  We do this because there are infinite stochastic representations and the MI expressions we consider are not convex in the parameters of $p(Z | S)$, making searching over these representations difficult.
Given each state representation, we compute the value of the MI objective as well as the optimal value function using exact value iteration.
In these examples, we assume that the policy distribution is uniform, and that the environment dynamics are deterministic.
Since we consider the infinite horizon setting, we use the steady-state state occupancy in our calculations.

The deep RL experiments with the catcher game are conducted as follows.
First, we use a uniform random policy to collect 50k transitions in the environment.
Next, each representation learning objective is maximized on this dataset.
To do so, images are resized to 64x64 pixels, normalized, and encoded with a convolutional encoder into a latent vector with dimension 256. The architecture of the encoder is 5 convolutional layers with ReLU activations.
We detail the way this latent vector is used for each algorithm below:

*Inverse information*: We interpret the latent embeddings of the images $s_t$ and $s_{t+1}$ as the parameters of Gaussian distributions $p(Z | s_t)$ and $p(Z | s_{t+1})$. We obtain a single sample from each of these two distributions, concatenate them and pass them through a single linear layer to predict the action. The objective we maximize is the cross-entropy of the predicted actions with the true actions, as in Agrawal et al. 2016 and Shelhamer et al. 2016. To prevent recovering the trivial solution of preserving all the information in the image, we add an information bottleneck to the image embeddings. We tune the Lagrange multiplier on this bottleneck such that the action prediction loss remains the same value as when trained without the bottleneck. This approximates the objective $min_{\phi} I(Z; S) s.t. I_{inv} = max I_{inv}$. To use the learned encoder for RL, we embed the image from the current timestep and take the mean of the predicted distribution as the state for the RL agent.

*State-only information*: We follow the Noise Contrastive Estimation (NCE) approach presented in CPC (Oord et al. 2018). Denoting $z_t$ and $z_{t+1}$ as the latent embedding vectors from the convolutional encoders, we use a log-bilinear model as in CPC to compute the score: $f(z_t, z_{t+1}) = \exp (z_t^T W  z_{t+1})$ for the cross-entropy loss. We also experimented with an information bottleneck as described above, but found that it wasn’t needed to obtain insufficient representations. To use the learned encoder for RL, we embed the image from the current timestep and use this latent vector as the state for the RL agent.

*Forward information*: We follow the same NCE strategy as for state-only information, with the difference that we concatenate the action to $z_t$ before computing the contrastive loss.

The RL agent is trained using the Soft Actor-Critic algorithm, modified slightly for the discrete action distribution (the Q-function outputs Q-values for all actions rather than taking action as input, the policy outputs the action distribution rather than parameters of a distribution, and we can directly compute the expectation in the critic loss rather than sampling). The policy and critic networks consist of 2 hidden linear layers of 200 units each. We use ReLU activations.

---

### Public Comment · ~Rohin_Shah1 · 2020-11-16
**Issues with the proofs**

I really like the premise of this paper: to investigate whether the incentives given by various unsupervised learning objectives are sufficient for learning representations that support an optimal control policy. The main paper is very clear, explains its points well, proves that the forward information objective is sufficient, shows simple counterexamples for the sufficiency of the other two objectives, and demonstrates the qualitative type of failures that can arise given this lack of sufficiency.

Unfortunately, after looking at the proofs, it turns out that the proof of sufficiency of forward information is incorrect: Lemma 2 is false, which is used in the proof of the theorem proving sufficiency of forward information.

In the proof, the incorrect step is when it is asserted that H(A+B) - H(A|Y) - H(B|Y) = I(Y; A+B); this should be <=, not =, and once this is done the rest of the proof no longer follows. (The same thing happens with Y replaced by X elsewhere.)

The statement of the lemma itself is incorrect -- here is a counterexample:

A = Uniform({1, 2, 9001, 9002})

B = 2*N + A, with N sampled from Uniform({1001, 1002, ... 2000})

X = Indicator(is A odd?)

Y = Indicator(is A larger than 5000?)

Here, knowing A obviously reveals the answers to X and Y, and B is constructed such that these answers can still be inferred. (A is odd iff B is odd; A is larger than 5000 iff B is larger than 5000.)

A + B = 2*(N+A) which is always even, so it is nearly impossible to recover whether or not A was odd, but the magnitude of A can still be easily recovered. This is a counterexample, since we have:

I(X, A) = 1 bit

I(X, B) = 1 bit

I(X, A+B) = about 0

I(Y, A) = 1 bit

I(Y, B) = 1 bit

I(Y, A+B) = 1 bit

My guess is that in fact the main theorem is also false, and that forward information is not sufficient for control. You need to have forward information as well as reward information, that is, maximize I(R_t; Z_t) explicitly in addition to I(Z_{t+k} ; Z_t, A_t). Consider the following MDP:

```
    /-D
 /-B--E
A
 \-C--F
    \-G
```

There are two actions, 1 and 2. In state A, both actions have a 50% chance of moving to B and 50% chance of moving to C. In state B, action 1 goes to D, and action 2 goes to E. In state C, action 1 goes to F, and action 2 goes to G. In states D, E, F, G, all actions are self-loops.

I believe (though haven’t checked carefully) that in this MDP, you can have a state representation that aliases all of the following pairs: {B, C}, {D, F}, {E, G}, while still maximizing forward information. This is insufficient for control with e.g. the reward that is 1 on E and F, and 0 everywhere else. Adding in I(R_t; Z_t) into the objective would fix this issue.

Typos:

In the caption of Figure 2, $J_{inv}$ should be $J_{state}$.

---

> ### Comment · AnonReviewer4 · 2020-11-19
> **Glad to see this being raised.**
>
> This corroborates with the third point in my minor comments. I agree that the premise of the paper is interesting but the proof seems problematic.

---

> ### Author Response · Authors · 2020-11-19
> **We have updated the proof; the proposed counterexample is flawed**
>
> Thank you very much for your interest and careful reading of our paper!
> Regarding Lemma 2, we agree that as stated in the paper it is incorrect. We have revised the proof of Proposition 1 and have updated the paper. We’d be happy to know if you spot any remaining issues.
>
> We believe there is an error in the proposed counterexample to the sufficiency of forward information.
> First, note that since we are in the infinite horizon case, if the states D, E, F, G are sink states then the state-occupancy distribution would not have full support. In our paper we assume that the policy distribution and state marginal distribution have full support. (If they didn’t, then there could be a high-reward state outside the support which is not represented because representing it would not increase MI. This is a fairly weak assumption; it would be quite difficult to propose a representation learning method that could possibly learn to represent states outside the support.) Therefore we will assume that 1 action from each of these states leads deterministically back to A. The state occupancy distribution $\rho$ satisfies $p(S’|S) \rho(S) = \rho(S)$. The state occupancy distribution is then $\frac{1}{4}$ for state A and $\frac{1}{8}$ for every other state.
>
> Now we calculate $I(S_{t+1}; S_t, A_t) = H(S_{t+1}) - H(S_{t+1} | S_t, A_t)$.
> $H(S_t) = \frac{1}{4} \log(4) + 6\frac{1}{8}\log(8) = \frac{1}{2}\log(2) + \frac{9}{4}\log(2) = \frac{11}{4}\log(2)$
> $H(S_{t+1} | S_t, A_t) = 2\frac{1}{2}\frac{1}{2}\frac{1}{4}\log(2) = \frac{1}{8}\log(2)$
> $I(S_{t+1}; S_t, A_t) = \frac{11}{4}\log(2) - \frac{1}{8}\log(2) = \frac{21}{8}\log(2)$
> To calculate $I(Z_{t+1}; Z_t, A_t)$ for the representation suggested (aliasing B with C, D with F and E with G), we first calculate the state occupancy, which is uniform.
> $H(Z_t) = -4\frac{1}{4}\log(\frac{1}{4}) = \log(4)$.
> The dynamics distribution now has zero entropy since all transitions are now deterministic.
> $H(Z_{t+1} | Z_t, A_t) = 0$
> $I(Z_{t+1}; Z_t, A_t) = \log(4) = 2\log(2)$
> Comparing the calculations for the $S$ and $Z$ representations: $2\log(2) <  \frac{21}{8}\log(2)$.
>
> The intuition for the sufficiency of forward information is that it not only seeks to minimize the entropy of the dynamics $H(Z_{t+1} | Z_t, A_t)$ but also to maximize the marginal state entropy $H(Z_t)$, which the counterexample doesn’t consider.

---

> > ### Comment · ~Rohin_Shah1 · 2021-01-14
> > **Thanks**
> >
> > Thanks for the reply; I hadn't realized that you made the assumption that the state-occupancy distribution would have full support; I agree my counterexample doesn't work then.
> >
> > I still like this paper a lot; I hope it gets published in a future conference :)
> >
> > (I would have commented sooner but unfortunately the discussion period meant that I couldn't write new comments.)

---

### Author Response · Authors · 2020-11-25
**Summary of updates**

Dear all reviewers and AC,

We would like to summarize the updates we have made to the paper thanks to the helpful feedback of the reviews.

*Theory*

(1) Updated the proof of Proposition 1; to our knowledge it is now correct.

(2) Updated the counterexample for state-only information which previously had a mistake. Please see the reply to R4 for a step-by-step example of how we compute the MI in these examples.

*Experiments*

(1) Added the result for $J_{state}$ to the catcher experiment, and the $J_{inv}$ result to the catcher-grip experiment (see Figure 5), as requested by R1.

(2) Added an experiment predicting Q* from each learned representation (see Appendix 8.3), as requested by R2.

(3) Added experiments with background visual distractors to increase visual complexity (see Appendix 8.4) to provide a result on an environment where representation learning provides drastic improvement over end-to-end training, as requested by R5.

(4) Added experimental details for didactic and deep RL experiments (see Appendix 8.2), as requested by R5.

Thank you for your reviews and responses throughout the rebuttal period. Your feedback has been very valuable in improving the work!

---

### Decision · Program_Chairs · 2021-01-07
**Final Decision**

**Decision:**

Reject

**Comment:**

**Overview**:
The paper tries to answer which mutual information (MI) objective is sufficient  for representation learning (repL) in reinforcement learning (RL). Three common objectives are considered: forward, state, and inverse. The paper shows that only the forward objective is sufficient for learning, i.e., sufficient for learning of optimal policy/value function. The authors also demonstrate this phenomena using empirical experiments.

**Quality, Clarity, Originality and Significance**:
All the reviewers believe this paper is novel in terms of methodology, i.e., evaluate the sufficiency of the repL in terms of down stream tasks. However, there is a lack of clarity in the experiment sections. The authors have provided more details in the rebuttal phase. The reviewers also have concerns that this paper may be too far from typical experimental settings to have a real impact on the field. An unofficial review pointed out there is a mistake in the proof of the paper. The authors later also confirmed the flaw and claimed it is fixed.

**Recommendation**:
The paper is indeed interesting and novel. However, the impact to the practice community might not be significant. That being said, the paper should warrant publication eventually. However, the authors changed large amount of text about the proofs before and after rebuttal, which also introduced some additional typos, confusions, and also technique sloppiness or flaws. The reviewers are concerned about this. Overall I believe that the paper is not in a state to be published yet.